# Compassionate Care: A Qualitative Exploration of Nurses' Inner Resources in the Face of Burnout

Sarah-Louise d'Auvergne Flowers [1,2], Mireia Guillén-Solà [1,2,*], Noemí Sansó [1,2] and Laura Galiana [3]

1 Department of Nursing and Physiotherapy, University of Balearic Islands, 07122 Palma, Spain; s.flowers@uib.cat (S.-L.d.F.); noemi.sanso@uib.es (N.S.)
2 Balearic Islands Health Research Institute (IDISBA), 07120 Palma, Spain
3 Department of Methodology for the Behavioral Sciences, University of Valencia, 46010 Valencia, Spain; laura.galiana@uv.es
* Correspondence: mireia.guillen@uib.eu

**Abstract:** There is a universal shortage of nurses, with a current needs-based shortage of 5.9 million. This is not solely a recruitment issue but one of retention, triggered by high levels of work-induced stress, burnout, and reports of low job satisfaction resulting in poor care delivery. Some of the health repercussions on nurses include anxiety, insomnia, depression, migraines, irritability, absenteeism, and sometimes alcoholism and drug abuse. To tackle some of these costly issues, a qualitative exploration into how inner resources is used by nurses to cope with stress at different points of their careers is proposed. Through the lens of grounded theory, semi-structured interviews will be carried out with two distinct sets of participants: (1) Student nurses registered at the University of the Illes Baleares between 2022–2025. (2) Experienced nurses on the Balearic nursing register. Interviews will be coded and then analysed using Atlas.ti. Expected results will inform curriculum improvements that will benefit the well-being of (student) nurses, from the outset of their training, pre-empting potential psycho-social risks before they arise in the workplace. This is vital as it addresses nurses' mental health as well as chronic issues of retention and absenteeism.

**Keywords:** compassion; burnout; inner resources; student nurse; moral distress; work conditions

## 1. Introduction

It is of no surprise to read that nursing is a highly stressful profession, with much of the literature reporting elevated levels of burnout [1–4], work-induced stress [5–7], compassion fatigue [4,8], poor mental health of professionals [3,7,9], and consequent high drop-out figures [10–12]. Burnout is defined as a syndrome that results from chronic work-induced stress that has not been properly managed, characterised by exhaustion, mental distance, and negative attitude towards the job as well as reduced efficacy [13]. It is interesting to note that burnout has recently been added to the World Health Organisation (WHO) International Classification of Diseases (ICD-11) [13] and is often cited as a primary cause for nurses leaving the profession [3,6,7,14]. It is predicted that there will be a global shortfall of 9 million nurses by 2030 [15], resulting in an even higher work burden for those who remain. This situation has been exacerbated by the global COVID-19 pandemic, which has left an already overloaded workforce under further pressures [9]. Although COVID-19 is not the focus of this study, it is important to note the significant impact it had on nurses, as well as highlighting gaps in health policies important to nurse retention [16–18]. The challenging issues faced by nurses have also been highlighted in a macro-inquest undertaken by the Spanish Nursing body, Consejería General de Enfermería (CGE) in 2022, which declared that nurses were at particularly high risk of occupational induced stress and burnout [14]. Results showed that 98.7% of nurses felt politically neglected, 85% felt that their mental health had suffered greatly, with half of those interviewed having considered leaving the profession [14]. This demonstrates the severity of issues of retention and job satisfaction

within the nursing profession. Although the CGE results must be considered in light of the COVID-19 pandemic, they serve to illustrate a chronic, underlying, long-term problem [12].

Burnout is not new to nursing [19], as the literature dating back to the 1970s shows [20,21]. Indeed, an ever-growing body of research shows that burnout and compassion fatigue influence nursing outcomes in a plethora of negative ways [3,5,22–25]. Although inter-linked, these negative factors on nursing can be divided into three categories: personal, social, and economic.

The personal impact of burnout on nurses takes both a physical and psychological toll. Some of the repercussions of this stress noted within recent literature include insomnia, depression, anxiety, irritability, migraines, chronic health flare-ups, poor job satisfaction, increased absenteeism and sometimes alcoholism, drug abuse, and relationship problems [26–33]. Once nurses are exhausted and are struggling with their mental health, it is understandable that these components can impact on their care delivery, resulting in suboptimal standards and burnout [22].

This leads to the social impact of burnout. Negative attitudes, cynicism, and poor efficacy have consequences with regard to the quality of patient care. Burnout can be linked to reduced patient safety and satisfaction, due to increased clinical errors because of exhaustion and poor connection to patients [34–37].

This, in turn, has an economic impact, not only because of the costs generated by medical errors [4,6,10] but also because of absenteeism and the agency costs to cover nursing shifts. This puts even more pressure on remaining colleagues, who often have to take on the work of those who are absent [6,7,12].

Considering their high stress work environments, it is of interest to explore how nurses are impacted by it and how, in turn, their capacity to deliver compassionate care is affected [7], not only negatively but potentially positively as well. Compassion can be defined as "a virtuous response that seeks to alleviate a person's suffering and needs through relational understanding and action" [38]. Compassion is considered to be a foundational element of nursing [39–41] and is highly valued by patients [42], yet it is one of the first elements to be whittled away by the heavy burden of work-induced stress [43,44].

As can be seen, a great deal of the existent literature addresses compassion within nursing, as well as the negative factors rife within the field, such as compassion, fatigue, and burn out [29–32]. However, few look at a more *positive framing* of how nurses' inner resources can result in the action of compassionate care and ultimately higher job satisfaction [22,44]. In contrast to compassion fatigue, compassion satisfaction explores the gratification and pleasure that can be achieved from helping others, enhancing quality of life [44,45]. There is emerging evidence that compassion may act as a protector of this professional quality of life and may even limit the effects of burnout [25], something this study aims to explore further. This is important, as keeping professionals compassionate could result in good quality patient care as well as happier nurses [22,25].

Another area of significant interest, especially in consideration of retention issues in an already short-staffed workforce, is how student nurses and their capacity for compassionate care delivery is affected by high stress work environments and potential burnout. This is particularly relevant as students are the future of the profession. Recent concerns have been raised regarding few recruits, poor retention [46], and an ageing workforce, with many qualified nurses about to retire [1,12,14,47], thus reiterating the importance of caring for current and future nursing students. Although less apparent in the literature, a review shows that nursing students also struggle with burnout [48–51], not only due to stressful work/placement environments, but as a result of difficult transitions from the student education setting to placement, an experience students of most other professions do not experience [50]. In many ways, student nurses do not reflect the general student population due to a high proportion of female students—70% of global health workers are women compared to an average 41% in other sectors [15], with the European proportion of female nurses sitting at 89% [52]. Such statistics are coupled with the fact that nursing boasts higher-than-average mature students [12,53], which has further implications for their studies, as

many of these students will be trying to balance external factors, such as family duties with their studies, adding to the stresses of the workplace that can lead to burnout. Abram and Jacobowitz [51] suggest that burnout in students has less to do with placement and more to do with external factors. There is a sparsity in the literature of a deeper understanding of how these factors are experienced and managed by student nurses, through the positive framing of the use of their internal resources.

It would therefore be of interest to explore how exposure to the realities of the profession affect students as they undergo their training; what inner resources do they use to combat burnout [47,54] and are these different to nurses with more experience in the field? Here, inner resources refer to the five concepts identified by McGahie [55] that map the core of compassionate care amongst health care professionals. They are *cognitive resources*, such as optimism, resilience, efficacy, and concept of self; *affective resources*, such as positivity, adaptability, cooperation, and playfulness; *moral resources* exemplified by code of ethics, hope, and perceptions of a bigger picture; and *awareness of self and others*, demonstrated by reflection, adaptation, and positive interactions [22,55].

After an extensive literature review, some research gaps were identified, which we hope to address in this study. Within the Spanish literature, there have been several quantitative investigations on compassionate care delivery and what the influential factors for burnout are [56,57]. There are some studies on inner resources [22,58] and the use of self-care approaches to help nurses manage burnout [59]. Most of these studies focus on more quantifiable markers that have been very useful for establishing what some of the issues confronted by nurses actually are, as well as providing an insight into the numbers of professionals dealing with these challenges. Although some qualitative studies have taken place in Spain with regard to nurses and work-induced stress [60–62], there appears to be a research gap, and the paucity of empirical literature around the topic from a holistic perspective, with little light shed on how nurses' use internal resources (rather than what they are) to positively influence compassionate care. There has also been comparatively little qualitative research into a more positive angle on nurses' resilience and into how nurses choose to frame their feelings or how these lived experiences evolve and influence the quality of their care delivery, both to patients and themselves. Further, few studies exist (of which even fewer are qualitative) amongst the student nursing population in Spain. It is crucial to protect new recruits and future nurses, especially considering the risks they will potentially face in a profession rife with burnout [47,50].

So, instead of re-emphasising what has already been achieved, this qualitative study aims to address these research gaps, by discovering the positive impact of using inner resources, in order to compliment existent literature, by unpacking some of the more subtle issues in the minds and "on the ground" through the lens of a grounded theory inquiry.

## 2. Objectives

### 2.1. Main Objective

The main objective of this study is to obtain a detailed insight into how nurses use their experience and inner resources at different points of their careers and how this in turn affects their ability to deliver compassionate care in the face of high stress environments and burnout. This will be explored via the evolution of their perceptions, the utilisation of inner resources, and the use of compassion as a potentially positive resource to protect professional satisfaction.

### 2.2. Secondary Objectives

- Explore the inner resources determined by McGahie's model [55], as well as other emerging themes [63,64], in order to develop an explanatory model of the influence of inner resources on compassionate care delivery [64–66];
- Examine perceptions of compassionate care at different stages of nursing careers (students nurses and nurses with experience);

- Establish if and how these social processes evolve over time, with exposure to stressful environments and experiences in the workplace [67];
- Frame compassion positively as a potential tool to protect professional satisfaction and consequently professional (compassionate) care delivery [22,68].

**3. Methods**

*3.1. Study Design*

This is to be a qualitative study that will use grounded theory as its chosen approach. This is a research method concerned with the generation of theory, which is "grounded" in data that have been systematically collected and analysed [69]. Key characteristics include simultaneous data collection and analysis with one informing the other. Data collection is cyclical and reflective and is grouped into concepts, categories, and themes in a process influenced by the simultaneous development of those concepts, categories, and themes [70]. What is important here is the principle of constant comparison, such as the process of noting issues of interest in data and comparing them to other examples to identify similarities and differences [64]. Strauss and Corbin [71] suggest using the "six Cs" of social process when undertaking this process: cause, context, contingency, consequence, covariance, and conditions. The outcome is to create an explanatory theory of basic social processes studied within the environment in which they take place [65].

Grounded theory was chosen for this study in particular because it aims to examine and develop an explanatory theory of how social process is affected, in this case how nurses deal with burnout, by a specific context—nurses within their work environment. Previous investigations regarding compassion and nurses' inner resources in Spain have sought to answer these questions by measuring quantifiable biomedical markers, leaving a paucity of empirical literature around the topic from a holistic perspective. The use of a grounded theory approach offers new insights, data, and an in-depth exploration of this phenomenon to compliment the existent quantifiable data.

*3.2. Time Period*

The study will be developed as part of a doctoral thesis between January 2023 to December 2025. Ethics approval was granted by the university in May 2023. Recruitment, data collection, and analysis commenced in July 2023 and will continue simultaneously until saturation is met as per grounded theory methodology. The aim is to report on findings in December 2025.

*3.3. Participants*

Participants for this study are either student nurses registered at the Universitat de Illes Baleares, Spain, or qualified nurses on the Balearic nursing register (COIBA). Where possible, participant age and gender are to be targeted to reflect the general cohort. All participants will be over eighteen, due to being of university age or older.

Students will be approached from all four years groups of their training. This will be a snapshot of student/nurses at a particular point in their careers, a cohort rather than a longitudinal study. First years will be interviewed in their first semester before exposure to clinical practice, in order to capture pre-perceptions of compassionate care and potential burnout. Accordingly, interview questions will be more generalised, linked to base-line stress management. Second, third- and fourth-year students will all be interviewed at times that do not interfere with their studies and practice commitments. Interviews with these more advanced participants will be guided by their experiences on clinical placement, as well as exploring their views and understanding of their education.

Participants in the qualified nurse cohort will have between one year to thirty-five years of clinical experience. Interviews with these professionals will explore the concept of their inner resources and lived experiences in stressful work environments and will be guided by what they wish to place importance on.

Inclusion criteria:

- Nursing students registered at the Universitat de Illes Balears in their first through to fourth year of studies between 2022 and 2025;
- Qualified nurses with a minimum of one year's experience up to thirty-five years;
- Willingness to participate in the study, give voluntary consent, agree for anonymised transcripts to be used for further analysis and potential publication.

  Exclusion criteria:
- Other medical professionals that were not nurses;
- Unwillingness to comply with proceedings.

*3.4. Sample*

Purposeful sampling has been chosen for the initial stages of recruitment in order to ensure that participants are typical cases of the subject being investigated [65], in this case nurses facing burnout. This allows for in-depth knowledge to be gained from those experiencing this particular social process. This will be followed up by theoretical sampling, which is a key aspect of the grounded theory approach [63]. Recruitment continues until the sample finally represents all aspects that make up the theory of the data represented [65]. Participants are continuously recruited based on their different experiences of a phenomenon, allowing for multiple dimensions of the social process to be explored [64,72], so in this study, nurses at various points in their training and careers. Saturation, which is accumulative, will be met once a complete range of experienced constructs is represented by the data and no new theoretical insights are forthcoming [63,72]. Due to pursuing data rich information, rather than a large quantity of it, as per grounded theory, the aim for this study is to gain between 10–30 interviews [65,73,74].

*3.5. Data Collection*

Data will be collected in the form of one-to-one interviews with voluntary participants, who have agreed to sign a consent form. The lead researcher will oversee recruitment and undertake all interviews to ensure consistency; however, subsequent codes will later be reviewed by a second researcher to reduce bias. Interviews will be recorded (audio only), transcribed, and anonymised as per data protection protocol.

Students will be contacted at the university via email, the student "Moodle" platform, notice boards, or by making announcements at the end of classes. Professionals will be approached by email, virtual notice boards, the licensing body news bulletin, and via ward supervisors (see Supplementary Figure S1 and Document S1).

Interviews will be held in a room at the university, which is a neutral, private space with little noise so as to guarantee comfort and intimacy for the participant. They will last between 40 min and an hour.

A question guide will be used for the semi-structured interviews to ensure that the key themes are addressed (see Supplementary Document S1, for a complete list of detailed guiding questions, divided into themed blocks). However, as per qualitative protocol, the participant will steer the direction of the interviews with the issues they find to be most relevant [72,73]. The main blocks of questions to be considered are as follows:

- Welcome, purpose of study explained, opportunity to resolve doubts, and ice-breaker/ establish participant characteristics;
- Work conditions and professional quality of life;
- Inner resources, where: cognitive capacity, spirituality and morality, affect and awareness of self and others will be explored, as well as other resources suggested by participants;
- Compassionate care and what it means to the participant, and an exploration of it as a positive tool in the work place;
- Proposals for future improvement of education/support/mental health;
- Closing remarks and space for questions and comments.

The guide will be pilot tested with both students and professionals before mass recruitment is rolled out, to ensure that the questions are appropriate for participants at

different points of their training/exposure to stress. These will be more generalised for first years, increasing in detail with the amount of experience held by the participant.

*3.6. Data Analysis*

Audio recordings of interviews will be transcribed verbatim and saved in a Word file. These will then be anonymised to protect participants identity by creating unique codes that will reflect years of experience and gender only.

Textual data will then be uploaded to qualitative data analysis software, Atlas.ti Mac (version 23.2.1) in order for comprehensive readings and in-depth analysis of open-ended questions to be explored and themes extracted. Atlas.ti is particularly useful for qualitative data analysis, as it allows for codes to be applied to text, either by a deductive approach, via the creation of a code book (prepared before-hand with presupposed ideas from the literature review), or an inductive approach where data are analysed line by line, allowing for codes to be refined according to the raw data. The programme enables large amounts of data to be visually organised and edited by multiple team members, so reducing manual agreement. Content analysis will follow a grounded theory approach where data collection and analysis will occur simultaneously, allowing for emerging themes and issues to inform researcher actions at each step [64,72]. This analysis involves the break-down of interview transcripts, line by line, allowing for key phrases and words to be identified and moved into categories, known as open coding, and then into further subcategories known as axial coding [69,72]. This is repeated until theoretical saturation is reached. These codes, which will reflect the participants' own words, so keeping them grounded in the data [75], result in a theoretical framework that explains the phenomenon. This enables theory construction and the development of fresh concepts [70]. It is important to note here that this methodology does not aim to test hypothesis but instead to generate data and new perspectives, helping to explain a process, and not to test an existing theory [64].

## 4. Trustworthiness

*4.1. Rigour*

To ensure the trustworthiness of this study and that a rigorous approach will be taken to tackle the complexity of qualitative data, credibility, auditability/repeatability, and fittingness will be considered [75–77]. To do this with regard to data analysis, Chiovitti and Parvin's [75] eight methods will be used, whilst COREQ guidelines will be used to ensure quality reporting [78].

*4.2. Credibility*

Credibility will be enhanced by including varieties of data. Accordingly, data will be triangulated via theoretical sampling, allowing for different interpretations of the same phenomenon to be exposed, i.e., the use of interviews with students as well as professionals. Bias will be avoided by recording interviews and keeping a research journal to reflect on researcher interpretations and observations [77]. Emerging codes and categories will be reviewed by a secondary researcher, to minimise lead-researcher bias and ensure that similar conclusions are being drawn. This will be facilitated using Atlas.ti 8.4 software that allows for an ICA (inter-coder agreement) calculation to be made. This is important as not only will researcher codes be triangulated, but any discrepancies will be highlighted immediately, so that they may be addressed during research, instead of at the end.

*4.3. Social Desirability Bias*

Four areas will be considered to minimise possible social desirability bias [79,80] that may result from the sensitive topics discussed and the potentially vulnerable interview environment. Firstly, anonymity will be assured, so that participants know that all information given cannot be linked back to them, thus allowing them to speak freely with reduced concern for personal impression management [79]. The second caution will be "scene-setting" [73], where the researcher will clearly explain that there are no right or

wrong answers, that anecdotes and personal viewpoints are encouraged. The aim here is to create an environment where participants feel that perspectives can be collected but not judged, thus reducing the tendency towards social desirability bias. In addition, only limited information will be given of the overall project to avoid predisposing ideas. Finally, very careful wording will be used when asking questions, so as not to steer participant views to a researcher's desired outcome [73]. Although it may not be possible to dismiss the risk of social desirability bias completely, it is hoped that these measures, as well as researcher neutrality and discretion will minimise it.

### 4.4. Auditability/Repeatability

Specification of why and how participants are selected, as well as what criteria is to be built into the researcher's thinking as part of an auto-checking process, will demonstrate transparency and auditability for future research [75,77]. (See Table 1 for examples).

**Table 1.** Examples of the criteria and checks used during analysis.

| Questions Asked by Researcher |
| --- |
| What is happening in the data? |
| What does the action in the data represent? |
| Is the conceptual label/code part of the participant's vocabulary? |
| In what context is the code/action used? |
| Is the code related to another code? |
| Is the code encompassed by a broader code? |
| Are there codes that reflect a similar pattern? |

Guiding questions derived from Glasser and Strauss, Strauss and Corbin, Chiovitti and Parrvin, Chamarz and Thornberg [63,64,70,71,81].

### 4.5. Fittingness

The aim here is to explain the scope of the research and the type of theory expected to be induced, thereby explaining its "fittingness" to the wider academic debate as well as how the literature relates to each emerging category, again, linking the study to the wider field it hopes to contribute to. Theoretical ideas will also be checked against participants' own meanings of the phenomenon, by unpacking and probing into what participants mean by asking for explanations and examples in their own words, instead of researcher's own, thus reflecting the participant's voice in the study [75].

### 4.6. Reflexivity

The challenge of reflexivity will be addressed by maintaining memos, a post-interview comment sheet and a personal research journal throughout the process, thus recording ideas and illuminating preconceptions about emerging data that may affect how data are interpreted by the researcher [74,75,77]. A section that acknowledges the researchers background and prior knowledge will also be included as an act of transparency [74,75].

## 5. Ethical Considerations

To meet the best practice regulations, this protocol was evaluated and approved by the Research Ethics Committee of the Universitat de Illes Balears on the 21 May 2023 with reference number 324CER23.

Three main areas were considered to ensure that this project meets ethical standards: confidentiality, autonomy, and fairness.

### 5.1. Confidentiality

As per university policy and the Spanish Organic Law 3/2018, dated 5 December, for the protection of personal and digital data, all personal information will be anonymised

and coded, as well as all interviews being held in a private space. With regard to data management, all interview transcripts will be stored on an encrypted and password protected hard drive, to which only the research team will have access. An application for "Tractament de Datos" at the university's data repository has been registered (Code107) so that data will be temporarily and safely stored for five years. Data minimisation will also be considered with regard to audio recordings, so that personal identifiers, such as workplace or job title, are pseudomised, which means that no one could use any reasonably available means to identify participants from the data: for example, interview number/year of study (1, 2, 3, 4) or experience (E)/male or female (M or F)—010: 2F or 011: EM. Once anonymised, the original audio files will be kept in the university's secure repository, as per protocol. Any data transfers and correspondence will be conducted via university email or encrypted pen drive.

### 5.2. Autonomy

To promote participant autonomy, it will be made clear that participation is completely voluntary. Although the researcher will reach out to potential interviewees, it is the participants themselves who will get in touch with the researcher to confirm their desire to be involved. The researcher will then obtain two signed, informed consent forms, with the option to withdraw at any time should the participant decide (see annex for examples of participant information and consent forms).

### 5.3. Fairness

This recruitment approach is fair, as it is a direct approach to participants who match the required profile for this study, as well as being completely voluntary, as initially, it is the participant who contacts the researcher. A debrief will be offered to all participants should they require it, as the subject matter may raise difficult topics, regarding patient death, work-related stress, and reasons for burnout. The advantage of using a semi-structured format here is that direct questions about traumatic experiences can be avoided, This will allow participants the flexibility to narrate their own experiences and in this way minimise the risk of triggering stress. In the case of significant distress, the participant will be sign-posted to the relevant support body.

## 6. Expected Results

An overall expected result is to develop an explanatory theory that will allow for the design of interventions that will support enhancements to nurse training programmes, so as to improve mental health outcomes for students and future professionals. This will feed into the development of good evidence-based practice, which will keep the field of nursing not only up to date but also in touch with those who make it happen.

### 6.1. Potential Limitations and Benefits

Potential challenges of this study revolve around the nature of the topic being explored: nurses who feel burnt out and being time-stretched may be less likely to volunteer their own time to participate in a work-related study. To address this, a strategy to boost dissemination of recruitment information will be adopted. As well as placing announcements on social media (that of the university, nursing department, research group, and lead investigator), posters will be placed in hospital staff rooms, as well as at the nursing licensing body training rooms and virtual notice board. It is hoped that by emphasising the importance of the goals to be achieved, facilitating the interview process as much as possible, and providing a convenient time and place for interviews, nurses will volunteer to participate.

There will be large amounts of qualitative data to process; to do this thoroughly, the abovementioned tools will be used to ensure that meticulous management and analysis take place. This is both the challenge and advantage of a grounded theory method—it will generate a considerable amount of rich data that can be used to inform good evidence-based practice.

*6.2. Anticipated Outcome*

Considering the inductive nature of grounded theory, where the aim is to produce an explanatory framework that integrates concepts into categories, it can be challenging to predict an outcome. The aim is to explain processes rather than to test existing theories [64,82]. It is the data collected that guide analysis and result in theory creation, leading to new discoveries (yet to be made), which can then inform policy, rather than to test the validity of existent ones [71]. That said, it is hoped that the results of this study will illuminate not only what but how inner resources and compassion, as a tool, are used by nurses in the face of high levels of stress to combat burn out.

We anticipate that the presence of nurse compassion will alleviate some of the negative implications that high levels of work-induced stress can bring and indeed may even have a role in protecting job satisfaction [9,22,44]. This in turn could result in better patient care, happier nurses, and higher levels of nurse retention—a benefit to all involved. Although we cannot be sure at this point which inner resources nurses use to foster compassion, we expect to see the aforementioned concepts proposed by McGahie for understanding compassion in health care professionals [22,55]: cognitive, affective, and moral resources, as well as awareness of self and others. We anticipate that individuals who have developed a combination of these internal resources will be more likely to foster compassion and so be more resilient in the face of burnout.

As discussed above, of the available Spanish qualitative literature on nursing and burnout, there is very little that tackles it from a more holistic perspective and even less that explores the positive impact of compassion on nurses' inner resources and how it can positively influence care delivery as well as job satisfaction. The comparison between nurses at varying stages of their training and careers to be explored in this study will hopefully show how nurses' perspectives and use of internal resources evolve and adapt over time, as a result of exposure to the work environment. It is these coping mechanisms and their use that we wish to capture.

These expected results will provide data that will allow us to establish a theory built on participant experience, which in turn will facilitate a framework on which to construct positive changes for future nurse education, health policy, and nurses' well-being.

**Supplementary Materials:** The following supporting information can be downloaded at: https://www.mdpi.com/article/10.3390/nursrep14010006/s1, Figure S1: Recruitment poster; Document S1: Notice for student Communications Platform (Moodle).

**Author Contributions:** Conceptualisation, N.S., L.G. and S.-L.d.F.; methodology, M.G.-S. and S.-L.d.F.; writing—original draft preparation, S.-L.d.F.; writing—review and editing, S.-L.d.F., M.G.-S., L.G. and N.S.; supervision, M.G.-S., N.S. and L.G. project administration, S.-L.d.F., M.G.-S., N.S. and L.G. All authors have read and agreed to the published version of the manuscript.

**Funding:** This research received no external funding.

**Institutional Review Board Statement:** The study will be conducted in accordance with the Declaration of Helsinki and approved by the Institutional Ethics Committee of the Universitat de Illes Balears, reference number 324CER23, 21 May 2023.

**Informed Consent Statement:** Informed consent was obtained from all subjects involved in the study.

**Data Availability Statement:** The data presented in this study are available on a reasonable request from the corresponding author.

**Public Involvement Statement:** According to GRIPP2 (Guidance for Reporting Involvement of Patients and the Public) we aim to involve and engage a wide range of nursing students and working nurses to ensure that our research analyses are relevant to their expressed concerns of compassionate care.

**Guidelines and Standards Statement:** This manuscript was drafted using COREQ guidelines: Tong, A.; Sainsbury, P.; Craig, J. Consolidated criteria for reporting qualitative research: a 32-item checklist for interviews and focus groups. Int J Qual Health Care. 2007; 19(6): 349-357 for qualitative research.

**Conflicts of Interest:** The authors declare no conflicts of interest.

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
