# Peer review of "Compassionate Care: A Qualitative Exploration of Nurses’ Inner Resources in the Face of Burnout"

_nursrep, doi:10.3390/nursrep14010006_

Round 1

Reviewer 1 Report

Comments and Suggestions for Authors

The protocol was prepared very carefully. The topic is important, more research is needed in this area. The authors searched for quality literature, but there are errors in the numbering. Source 16 has two numbers, and subsequently other sources are incorrectly numbered.

Reviewer 2 Report

Comments and Suggestions for Authors

Comments and Suggestions for Authors

Dear authors, I am pleased to review this well written protocol which is part of a doctoral thesis. Although there is a great deal of data in the existent literature that explores this topic, your research could provide good evidence- based practice for the future.

Few issues need to be addressed:

 Introduction section:  Line 37- you included the acronym “CGE” without providing an explanation in the text

 Method section: 3.5 Data collection:  will the data be collected by the same person for all the interviews? This should be mentioned in the text.

I wish you all the best with your research!

Reviewer 3 Report

Comments and Suggestions for Authors

Thank you for the opportunity to read and review this manuscript. This manuscript presents a protocol of a qualitative study based on the Grounded Theory to explore nurses’ inner resources in dealing with burnout and delivering compassionate care.

In my opinion the topic is important and interesting. The methodology is appropriate for the study aims. However, there are several points I would suggest the authors to consider for revision, which are listed below.

1. In the abstract the authors stated that ‘semi-structured interviews and discussion groups will be carried out with two distinct sets of participants’ (line 16 - 17); however I did not find any information regarding the discussion groups throughout the manuscript. Please could the authors clarify this issue?

2. In reporting the protocol of a qualitative study using interviews, it would be essential to provide sufficient information on the content of the interviews to ensure its informativeness and reproducibility. In the Data Collection section (page 5) the authors only provided a very brief summary of the interview questions. I would encourage the authors to provide here a little bit more details of the interviews, which may include the topics/themes of each question block, content to investigate and examples of questions, ideally presented in a table.

3. The subheadings of section 6 and 6.2 are duplicate (both as ‘expected results’) and should be rephrased.

4. In this study participants are expected to talk about their working experience/stress/environment/burnout in a one-to-one interview; these may involve sensitive topics that can be particularly vulnerable to social desirability bias. I would encourage the authors to make more discussions on how this kind of potential bias will be minimized.

Comments on the Quality of English Language

The manuscript is overall well-written and easy to follow; however, a proofreading is recommended as there are grammatical errors in the text.

Reviewer 4 Report

Comments and Suggestions for Authors

Thank you for the opportunity to review this paper.  This paper describes a study that has yet to be completed.  The topic of nurse burnout has emerged as an issue in keeping the workforce.  There two elements that I couldn't find in the background and they are impact since COVID and technology.  These two elements have been significant in this discussion.  The plan is to use software for analysis and this can reduce the need for agreement.  This could be mentioned in the analysis or the plan for agreement.   Thank you.
